# Prognostic Values of Gene Copy Number Alterations in Prostate Cancer

**DOI:** 10.3390/genes14050956

**Published:** 2023-04-22

**Authors:** Abdulaziz Alfahed, Henry Okuchukwu Ebili, Nasser Eissa Almoammar, Glowi Alasiri, Osama A. AlKhamees, Jehad A. Aldali, Ayoub Al Othaim, Zaki H. Hakami, Abdulhadi M. Abdulwahed, Hisham Ali Waggiallah

**Affiliations:** 1Department of Medical Laboratory Sciences, College of Applied Medical Sciences, Prince Sattam Bin Abdulaziz University, Alkharj 11942, Saudi Arabia; 2Department of Morbid Anatomy and Histopathology, Olabisi Onabanjo University, Ago-Iwoye P.M.B. 2002, Nigeria; 3Department of Biochemistry, College of Medicine, Imam Mohammad Ibn Saud University, Riyadh 13317, Saudi Arabia; 4Department of Pharmacology, College of Medicine, Imam Mohammad Ibn Saud Islamic University (IMSIU), Riyadh 13317, Saudi Arabia; 5Department of Pathology, College of Medicine, Imam Mohammad Ibn Saud Islamic University (IMSIU), Riyadh 13317, Saudi Arabia; 6Department of Medical Laboratories, College of Applied Medical Sciences, Majmaah University, Al-Majmaah 11952, Saudi Arabia; 7Medical Laboratory Technology Department, College of Applied Medical Sciences, Jazan University, Jazan 82817, Saudi Arabia; 8Department of Clinical Laboratory Sciences, College of Applied Medical Sciences, King Saud University, Riyadh 11362, Saudi Arabia

**Keywords:** prostate cancer, gene copy number alterations, localised disease, advanced disease, risk stratification, progression-free survival

## Abstract

Whilst risk prediction for individual prostate cancer (PCa) cases is of a high priority, the current risk stratification indices for PCa management have severe limitations. This study aimed to identify gene copy number alterations (CNAs) with prognostic values and to determine if any combination of gene CNAs could have risk stratification potentials. Clinical and genomic data of 500 PCa cases from the Cancer Genome Atlas stable were retrieved from the Genomic Data Commons and cBioPortal databases. The CNA statuses of a total of 52 genetic markers, including 21 novel markers and 31 previously identified potential prognostic markers, were tested for prognostic significance. The CNA statuses of a total of 51/52 genetic markers were significantly associated with advanced disease at an odds ratio threshold of ≥1.5 or ≤0.667. Moreover, a Kaplan–Meier test identified 27/52 marker CNAs which correlated with disease progression. A Cox Regression analysis showed that the amplification of *MIR602* and deletions of *MIR602*, *ZNF267*, *MROH1, PARP8*, and *HCN1* correlated with a progression-free survival independent of the disease stage and Gleason prognostic group grade. Furthermore, a binary logistic regression analysis identified twenty-two panels of markers with risk stratification potentials. The best model of 7/52 genetic CNAs, which included the *SPOP* alteration, *SPP1* alteration, *CCND1* amplification, *PTEN* deletion, *CDKN1B* deletion, *PARP8* deletion, and *NKX3.1* deletion, stratified the PCa cases into a localised and advanced disease with an accuracy of 70.0%, sensitivity of 85.4%, specificity of 44.9%, positive predictive value of 71.67%, and negative predictive value of 65.35%. This study validated prognostic gene level CNAs identified in previous studies, as well as identified new genetic markers with CNAs that could potentially impact risk stratification in PCa.

## 1. Introduction

According to the GLOBOCAN 2020 Cancer Statistics, prostate cancer (PCa) is the 4th most common type of cancer diagnosed worldwide in 2020 after female breast, lung, and colorectal cancers [1]. It is also the most common cancer diagnosed in men, the 8th most common cause of cancer deaths in 2020, and the 5th leading cause of cancer deaths in men [1]. Currently, risk prediction for individual cases is highly prioritised in PCa management [2]. While most cases behave indolently and most patients die with their tumours, a substantial fraction of patients develops an aggressive disease, which requires radical therapy. However, because it is difficult for urologists to adequately predict the disease progression for individual cases, many PCa patients may receive aggressive management strategies, which are associated with reduced quality of life post-procedure [2,3,4]. It is currently difficult to differentiate indolent from aggressive cancers based on clinicopathological parameters alone [5,6,7]. The current best prognostic index—the Gleason grading of histopathological samples—has limitations in predicting the clinical behaviour of individual tumours: interobserver variation is high, grading scores for small diagnostic needle biopsies differ significantly from that of a prostatectomy specimen for each individual due to sampling problems, and morphologically identical samples may exhibit different clinical behaviours [5,6,7,8]. Over the years, many studies have investigated the molecular pathology of PCa [2,9,10,11,12]. However, the relationship between molecular alterations and clinicopathological indices of PCa is not yet fully understood [2,9,10,11,12]. Many studies have shown that gene CNAs are more important than somatic mutations [9,11,12]. However, no single molecular marker has been validated for risk stratification in this disease. The molecular alterations that can predict clinical aggressiveness and metastatic potentials of individual cancers are not fully established. An understanding of the molecular alterations that drive cancer progression and aggressiveness is essential to formulating the most appropriate management strategies. In this study, we explored the TCGA PCa cohort to (i) identify genes with CNA statuses that can potentially predict the clinical progression of PCa and (ii) determine whether any combination of genetic markers can stratify PCa cases into early- and late-stage diseases with any degree of accuracy. 

## 2. Materials and Methods

### 2.1. Study Cohort

This study analysed the clinicopathological, copy number segment, and gene copy number data of 500 PCa cases. All the genomic data were generated by TCGA Network’s Pan-Cancer Atlas initiative and deposited in the National Cancer Institute’s Genomic Data Commons (GDC) repository (www.portal.gdc.cancer.gov/repository, accessed on 23 November 2022) and the cbioportal (www.cbioportal.org, accessed on 23 November 2022) from where they were downloaded and analysed. 

### 2.2. Clinical and Genomic Data

Level 3 copy number segment data of the 500 TCGA PCa cases were downloaded from the GDC repository while gene copy number and gene fusion data were retrieved from the cbioportal database. The clinicopathological data of the cohort was also obtained from both repositories. The copy number segment and gene level copy number data were generated by DNA SNP microarray. All the data were open-access and were freely obtained from the GDC and cbioportal repositories. 

### 2.3. Genomic Data Analyses

Data retrieval and analyses were accomplished using codes and scripts, which were written in simple Linux commands. See Appendix A.

### 2.4. Study Approach and Principles

Genetic markers shown from previous studies to have prognostic values were included in the study [2]. Moreover, potential prognostic genetic markers were retrieved from the breakpoints of recurrently altered chromosomal segments in the copy number segment data as follows: the genomic coordinates of recurrent (i.e., occurring in ≥5% of cases) and significantly altered (segment mean of ≤−0.3 or ≥0.3) chromosomal segments were retrieved from the TCGA masked copy number segments data for the PCa cohort; the genomic coordinates of the altered segments were then input in ENSEMBL BioMart tool to retrieve genetic markers from the altered chromosomal segments; then, the gene level copy numbers of all the prognostic genetic markers identified from the published literature, as well as genetic markers representative of those recovered from our genomic breakpoint analyses, were obtained from the cbioportal gene level copy number data for the TCGA PCa cohort. The prognostic significance of the copy number alterations of the identified markers was sought using the appropriate statistical tests.

The PCa cases were classified into a two-tier staging scheme by combining the AJCC clinical and pathological indices. In this two-tier scheme, “localised disease versus advanced disease”, the pathological indices were weighted more than the clinical indices. The pathological tumour stages T1-T2c were classified into “localised disease” in the absence of nodal (N1) or distant metastases (M1). Pathological T3a-4 was classed as “advanced disease” irrespective of nodal or distant metastasis. However, when the pathological tumour staging was not available for any case, the clinical staging was utilised for classifying that case; clinical T1-T2c cases without nodal or distant metastases were classed as “localised disease”, while T3a-4 cases were put in the “advanced disease” group irrespective of the nodal or distant metastases status. 

The prognostic grade groups, dichotomised prognostic grade groups (good prognostic grade group (prognostic grade groups 1–3) versus poor prognostic grade group (prognostic grade groups 4 and 5), age group, dichotomised age group (“young versus old”, using 60 years old as threshold), and Gleason scores were computed from the primary data for age at diagnosis, primary diagnosis, primary Gleason grade, secondary Gleason grade, combined primary and secondary Gleason grade, and Gleason scores and used for further analyses. 

### 2.5. Statistical Analyses

Clinical and molecular data were output in Comma Separated Values file (.csv) format to enable tabulation of clinical, copy number segment, and gene copy number data. All data were then transferred to SPSS version 24 as categorical variables (age group, ethnic group, disease stage (early versus late; pathological staging), Gleason’s grading, gene copy number, copy number segment, etc.) and continuous variables (age, Gleason’s score, etc.). All statistical analyses were performed in SPSS, and a *p*-value of <0.05 was regarded as significant. Correction for multiple testing was accomplished with the Seed Mapping online false discovery rate (FDR) calculator (https://www.sdmproject.com/utilities/?show=FDR, accessed on 15 December 2022) where applicable. Associations between two or more categorical variables were evaluated using the Chi-square (or Fisher’s) test. Odds ratios, which were retrieved from SPSS chi-square result tables and confirmed by manual calculations, were used to test the relationship between the individual gene CNA status on the one hand and the two-tier PCa staging and other prognostic indices on the other while binary logistic regression was used to determine the panel of the minimum number of genetic markers that could predict disease stages. Cox regression analysis and Kaplan–Meier tests were used to determine the relationship between copy number alterations of genetic markers and progression-free survival.

## 3. Result

### 3.1. Clinicopathological Characteristics of PCa Cohort

The clinicopathological indices comprising age, ethnicity/race, clinical stage, pathological stage, and Gleason’s primary and secondary grades of the 500 PCa cases are shown in Appendix A. The two-tier staging scheme, the Gleason grading indices, and the progression-free survival were tested as the prognostic indices for this PCa cohort. 

### 3.2. Copy Number Segments and Genetic Markers

A total of 12 significantly altered (segment mean of ≤−0.3 or ≥0.3) chromosomal segments bordered by recurrent breakpoints (present in ≥5% of the PCa cohort) were retrieved from the copy number segment data. These included chromosomal segments 3p11.1-3q29, 5q11.1-5q35.3, 7p11.2-7q36.3, 8p11.1-8q24.3, 9p11.1-9q34.3, 10p11.1-10q26.3, 12p11.1-12q24.33, 13q11-13q34, 16p11.2-16q24.3, 17p11.2-17q25.3, 18p11.21-18q23, and 21q11.2-21q22.3.

A total of 25 recurrent breakpoints flanking altered chromosomal segments were obtained from the chromosomal segment analysis, and from these, 264 genes were retrieved (see Appendix A). The CNAs of 21 genetic markers representative of the 264 genes were retrieved from the cbioportal gene copy number data (see Appendix A). The copy number of 31 markers of potential prognostic significance, which were identified from published literature, were also included in the analysis (Appendix A). Both deletion and amplification/duplication copy numbers were found for all the genetic markers except for *TP53,* which showed only deletions. For binary logistic regression analyses, all the genetic markers were coded as “Deletion versus No deletion”, “Duplication/Amplification versus No duplication/amplification”, and “Alteration versus No change” (deletion and duplication/amplification were coded together as an alteration). 

### 3.3. Gene Level Copy Number Alterations Are Associated with Disease Stage

A chi-square test with an odds ratio determination was utilised to establish whether the individual gene CNA status can stratify PCa cases into localised or advanced diseases. The results showed that 51/52 genetic markers, comprising 35 amplification statuses and 38 deletion statuses, showed a significant association with the disease stage at a *p*-value of <0.05. A total of 22 genetic markers showed a significant association with the disease stage for both their deletion and amplification statuses. The genetic alterations without significant associations with the disease stage included the TMPRSS2–ERG fusion, *NCOR1* amplification, and *ETV6* amplification. 

The odds ratio for advanced disease ranged from 0.608 (*SPP1* amplification) to 26.485 (*SPP1* alteration). Fifty-one genetic markers (51/52) showed significant associations with the disease stage at an odds ratio threshold of ≥1.50. These included 29 amplification and 38 deletion statuses. Sixteen (16/52) genetic markers had significant associations with the disease stage at an odds ratio of ≥1.5 for both amplification and deletion statuses. Six (6/52) genetic markers were significantly associated with the disease stage at an odds ratios of ≤0.667. These included *BCL2* amplification, *CCNE1* amplification, *CHD1* amplification, *CLIC4* amplification, *ETV4* amplification, and *SPP1* amplification. The latter CNAs were only present in advanced disease stages. 

The top 25% (or 75th percentile) genetic CNA with the highest odds ratios for advanced disease included *SPP1* alteration (i.e., combined deletions and amplifications), *SPP1* deletion, *ERG* amplification, *MCM3APAS1* amplification, *SPOP* amplification, *TMPRSS2* amplification, *HCN1* amplification, *PARP8* amplification, *ZNF267* amplification, *MIR602* amplification, *CDKN2A* amplification, *WNK1* amplification, *SPOP* alteration, *ZBTB16* amplification, *CCNE1* alteration, *MIR602* alteration, *CCND1* alteration, *SPOP* deletion, *HCN1* alteration, *PARP8* alteration, *CCND1* amplification, *AR* deletion, *MED12* deletion, *AR* alteration, *HCN1* deletion, *MED12* alteration, *CNTN4* amplification, *ANKRD20A11P* amplification, *COLEC12* amplification, *CDKN2A* alteration, and *PARP8* deletion. Common oncogenes and tumour suppressor genes whose CNA statuses were confirmed to have association with the disease stage at a *p*-value of <0.05 in this study included *CCND1, CCNE1, CDKN1B, CDKN2A, CHD1, MYC, PIK3CA, PTEN, RB1, FGFR1, BCL2*, and *TP53.* In addition, the PCa-specific genes confirmed to have significant associations with the disease stage included *AR, NCOR1, NCOR2, FOXA1, MED12, ERG, ETV4, ETV6,* and *TMPRSS2*. However, the TMPRSS2–ERG fusion did not show any significant association with the disease stage in this study. 

Furthermore, the CNA status of the following genes were confirmed to be associated with the prognosis in this study: *FOXO3, NCOA2, NKX3.1*, *ZBTB16*, *PMP22, CLU, CLIC4, SPOP*, and *SPP1* (Appendix A). 

The CNAs of all but one of the 31 potential prognostic markers retrieved from the published literature and 21 genetic markers obtained from our copy number chromosomal segment analyses showed associations with the disease stage in one or both forms. 

### 3.4. Survival Significance of Gene CNAs 

Kaplan–Meier analyses with a Log Rank test identified 31 features, including 3 clinicopathological features (disease stage, Gleason grade, and Gleason grade group) and 27 marker CNAs, which correlated with the disease progression in this PCa cohort (see Appendix A, only features with significant associations with progression-free survival shown). 

Multivariate analyses using Cox Regression analysis with correction for collinearity showed that *MIR602* amplification, *MIR602* deletion, *ZNF267* deletion, *PARP8* deletion, *MROH1* deletion, and *HCN1* deletion were associated with a reduced progression-free survival independent of the disease stage and Gleason prognostic group grade during a follow-up period of 10 years (see Table 1 and Figure 1).

Binary logistic regression analyses were performed to identify panels of genetic markers that can stratify the PCa cases into localised and advanced diseases. To reduce multicollinearity in the logistic regression analyses, the 51 genetic markers were combined into 9 groups of 19 markers, in which only the markers that map to different chromosomal regions were included in the same groups (see Appendix A). The multicollinearity diagnostic tests confirmed that the markers in each of the 9 groups did not have multicollinearity based on the following thresholds: tolerance = 0.25, variable inflation factors = 4, and condition index = 15. 

Each of the 51 genetic markers was input into the models as amplification, deletion, and alteration, up to a total of 154 variables, including the TMPRSS2–ERG fusion. Genetic markers with a non-significant contribution to the prediction of the disease stage (*p*-values ≥ 0.05) were excluded sequentially, starting with those with the highest *p*-values, until all the markers left in the model had a *p* value < 0.05. 

The analysis revealed twenty-three panels of marker CNAs with risk stratification potentials. The accuracy and sensitivity of the generated models ranged from 64.89–71.0% and 60.60–85.4%, respectively (see Appendix A). Based on the sensitivity rate of stratifying the cohort into localised and advanced diseases, the best fit logistic regression model derived contained 7/52 markers, including the *SPOP* alteration, *SPP1* alteration, *CCND1* amplification, *PTEN* deletion, *CDKN1B* deletion, *PARP8* deletion, and *NKX3.1* deletion. The model was statistically significant at X^2^ = 27.553 (*p* < 0.001), explained 28.0% (Negelkerke R^2^) of the variance among the cases, and accurately classified 70.0% of them into localised and advanced diseases. The Hosmer and Lemeshow test showed that the model did not have a poor fit at a *p*-value of 0.976. The sensitivity for the prediction of advanced disease was 85.4%, the specificity was 44.9%, the positive predictive value was 71.67%, and the negative predictive value was 65.35%. Of the genetic markers used in the model, the *SPP1* alteration displayed the highest odds for an advanced disease stage at 13.231 while *PTEN* deletion showed the lowest odd (2.032) (see Table 2).

## 4. Discussion

In this study, we retrieved the most altered chromosomal segments in the TCGA PCa cohort in order to obtain the most commonly altered gene copy numbers. The chromosomal segments retrieved from our copy number segment analyses overlapped significantly with the common chromosomal segments found by Camacho et al. in their study of 141 Cancer Research UK/International Cancer Genome Consortium cohorts of PCa [9] and by Williams et al. in their meta-analysis of somatic CNAs from 11 publications that examined 662 PCa patient samples [11]. As genomic re-arrangements are the predominant molecular alterations found in PCa [9,11,12], previous studies have attempted to use CNAs for prognostication. For example, Grist et al., Hieronymus et al. (2018), and Hieronymus et al. (2014) [13,14,15] explored the prognostic values of the total CNA burden (TCB), a measure that is comparable to the total mutation burden (TMB) used in prognostication in other cancers [16]. However, our study retrieved single gene CNAs from altered chromosomal segments and interrogated their outcome and disease-stage predictability. 

We confirmed the prognostic and risk stratification utility of previously studied gene level CNAs and other alterations [2,9,10,11,12,17,18,19,20,21,22,23,24,25,26,27,28,29,30,31,32,33,34,35,36,37,38,39,40,41,42,43,44,45,46,47,48,49,50,51,52,53,54,55,56,57,58,59,60,61,62,63,64,65,66,67]. For example, the CNAs of oncogenes and tumour suppressor genes, such as *PIK3CA, CHD1, MYC, PTEN, CCND1, CDKN1B, TP53, CCNE1, NKX3.1,* and *RB1,* have previously been shown to impact prognosis, tumour progression, and disease stage of PCa and other cancers [2,17,18,19,20,21,23,24,25,26,27,28,29,30,31,32,33,34,35,36,37]. Furthermore, molecular alterations of androgen receptor pathway genes, including *AR, MED12, FOXA1, NCOR1, NCOR2, SPOP, NCOA2,* and *ZBTB16,* have similarly been associated with different disease stages [2,22,33,37,38,48,49,50,51,52]. Besides *AR, NCOA2,* and *ZBTB16,* whose copy number changes were previously associated with late-stage diseases, the molecular alterations found in other members of the pathway were mainly somatic point mutations and altered expressions [2,18,30,34,35,45,46,47,48]. However, this study found that *MED12, FOXA1, NCOR1, NCOR2,* and *SPOP* exhibited copy number changes that were associated with disease stages. In addition, the gene expression patterns of *FGFR1, CLU, CLIC4,* and *PMP22* were used to stratify PCa with low Gleason scores into aggressive and indolent groups in the Irshad et al. study [39]. This study included the CNA status of *FGFR1, CLU, CLIC4,* and *PMP22* in the study since gene CNA generally correlates with gene expression levels [29] and found that the CNA status of *FGFR1, CLU, CLIC4,* and *PMP22* individually showed significant associations with PCa stage. In addition, *CLU* CNAs robustly fit into the logistic regression models 9, 10, 19, and 21 (Appendix A). *CDKN2A* deletion has been associated with prognosis in other cancers, such as gliomas [42], but not in PCa. This present study showed a *CDKN2A* deletion, an alternative mechanism of *CDKN2A* loss in PCa apart from the *CDKN2A* methylation revealed by the Ameri et al. study [40]. This study also showed that *CDKN2A* deletion, like *CDKN2A* methylation, is associated with prognosis. A *CDKN2A* deletion also robustly fit into the regression models 6, 16, and 18 (see Appendix A), evidence that the *CDKN2A* deletion can be combined in a panel of CNAs for PCa risk stratification. Somatic point mutation and upregulation of *BCL2* expression was associated with PCa progression by Catz and Johnson and Renner et al. [43,44]. In this study *BCL2* alterations were predominantly deletion events (27.52% of cases had deletion versus 1.44% with amplification, see Appendix A). However, both *BCL2* amplification and deletion were positively associated with disease stage in this study. Although *BCL2* deletion and reduced expression may appear counterintuitive for tumour cell survival and carcinogenesis, it should be noted that *BCL2*-deleted tumour cells may alternatively upregulate one or more members of the BCL2 family of anti-apoptotic genes, such as *MCL-1* and *BCL2L1*, to increase cell survival [45]. 

Our finding of an association between *ETV6* deletion and PCa stage is congruent with those of Liu [53] and Tsai et al. [54], who showed that *ETV6* downregulation is associated with invasion, migration, and metastatic phenotypes in PCa cell lines and xenograft models. In their models, which are supported by our data, *ETV6* acts as a tumour suppressor gene that negatively regulates the epithelial–mesenchymal transition and disease progression in PCa cells [53,54]. The *ETV6* deletion also fit into our predictive regression models 3, 7, 13, and 17 (Appendix A). The activities of *ETV4* in cell lines and xenograft models, on the other hand, are those of an oncogene, and the *ETV4* overexpression has been associated with metastasis [55,56,57]. This study found PCa stage associations for both the *ETV4* deletion and amplification. The amplification statuses of *TMPRSS2* and *ERG* were associated with the PCa stage in this present study, which is in concordance with previous studies [58,59,60]. However, TMPRSS2 fusion, ERG fusion, and the specific TMPRSS2–ERG fusion were not associated with a disease stage in this study. These findings are in congruence with the Toubaji et al. study, which found that an increased copy number of ERG, but not TMPRSS2–ERG fusion, predicted the outcome in PCa [61], and with the Albadine et al. study, which did not find any significant difference in rate of TMPRSS–ERG fusion between minute and non-minute PCa [62]. The failure of the *TMPRSS2–ERG* fusion to stratify PCa cases into early and late stages may be because the fusion is an early clonal event in prostate carcinogenesis [63,64,65,66]. In addition, the expression levels of *SPP1* have been associated with clinical stages, lymph node metastasis, and disease-free survival in PCa [67]. CNAs of *SPP1* were found to associate with disease stage, but not with survival, in this study. Furthermore, the *SPP1* alteration fit into all but one of our regression models. 

The novelty of this study is in the identification of 21 new gene CNAs with risk stratification potentials in PCa. A search of the literature revealed that the CNAs of these genes have not previously been documented to be related to PCa prognosis or risk stratification. We showed that *MIR602* amplification, *MIR602* deletion, *ZNF267* deletion, *PARP8* deletion, *MROH1* deletion, and *HCN1* deletion were associated with progression-free survival independent of the disease stage and prognostic group grade. 

*MIR602* maps to chromosome 9q34.3; encodes miR-602, a microRNA that promotes cell proliferation and metastases; and regulates the cell cycle in an oesophageal squamous cell carcinoma cell line model, in which it targets *FOXK2* [68]. *MIR602* is also involved in Hedgehog signalling in chondrocytes and has anti-apoptotic activities in hepatocellular carcinoma [69,70]. It is upregulated in gliomas and colon cancers [71,72]. *MIR602* overexpression has been associated with a poor prognosis in oesophageal squamous cell carcinoma and in glioblastoma multiforme [68,73] and a favourable outcome in pancreatic ductal adenocarcinoma [74]. The expression of *MIR602* is epigenetically regulated [68]. The present study showed that whilst both forms of *MIR602* alterations (i.e., amplification and deletion) were associated with disease progression, only *MIR602* amplification was significantly associated with disease stage and the regression models. Whilst *ZNF267* behaves as an oncogene in hepatocellular carcinoma, in acute lymphoblastic leukaemia, and in diffuse large B-cell lymphoma, in which its expression also predicts poor survival [75,76,77], this study found that *ZNF267* amplification is associated with the disease stage but not with progression-free survival. The *ZNF267* deletion, on the other hand, showed the reverse profile. Jiang et al., in a genome-wide association study of 5222 PCa patients, found that an inherited *MROH1* variant (or single nucleotide polymorphism, SNP) was one of the genetic factors that would warrant conversion of management strategies from active surveillance to treatment [78]. Zhang et al. found that *MROH1* was frequently altered as part of a chromosome 8q alteration in various cancers, including PCa [79]. Furthermore, Sharbatoghli et al. found that *MROH1* amplification is a predictive marker of drug response in ovarian cancer management [80]. Moreover, Harada et al. found that somatic mutations of *MROH1*, among other genes, were associated with metastatic gastric adenocarcinoma relative to the primary tumour [81]. In comparison, the present study found that an *MROH1* amplification was significantly more common in advanced disease than in localised tumours and can potentially stratify PCa into stages (regression model 17). In addition, the *MROH1* deletion was associated with a progressed disease in the TCGA PCa cohort. *PARP8* encodes PARP8, a member of the family of poly (ADP-ribose) polymerase with DNA repair, genome stability maintenance, and cellular homeostasis functions [82]. A specific function is yet to be described for *PARP8* per se, but PARP8 localises to the nuclear envelope for the majority of the cell cycle, except during mitosis when it localises to the centrosome and spindle poles [83]; hence, it may function in the maintenance of the mitotic spindle apparatus. It also functions in the cellular apoptotic pathway [84]. This present study found that *PARP8* deletion showed risk stratification potentials and robustly fit into 8 of our 22 regression models (see Appendix A). It was also inversely associated with progression-free survival. *HCN1* maps to chromosome 5p12 and encodes HCN1, a member of the hyperpolarization-activated cyclic nucleotide-gated channel (HCN1-HCN4) protein family whose expression was normally localised to the heart and nervous system [85,86,87]. *HCN1* SNPs have been associated with both risk and prognosis [86]. Phan et al. found that *HCN1* mRNA was overexpressed in prostate and other cancers, and this expression pattern was associated with an adverse overall survival in breast and colorectal cancers [87]. Two other members of the HCN family, *HCN2* and *HCN3*, were shown to be overexpressed in clinical breast cancer and to have adverse prognostic significance; they were also predictive markers for ivabradine response in in vitro breast cancer models [85]. No survival significance was stated for prostate cancer in that study. On the other hand, we found associations between *HCN1* amplification/deletion and disease stage in this PCa study. *HCN1* deletion also fit into 7 of our 22 regression models that stratified the PCa cohort into early- and late-stage diseases (see Appendix A) and independently predicted disease progression.

This study has explored the TCGA PCa cohort and identified new gene CNAs with risk-stratification potentials. The study has also determined that multiple gene CNAs can be combined in a regression model for risk stratification and disease progression prediction purposes. Our best logistic regression model showed a modest performance with respect to accuracy, sensitivity, specificity, and negative and positive predictive values for localised versus advanced diseases. Its component genetic CNAs also explained only 28.0% of the variance between localised and advanced disease stages even though we used a total of 154 variables. In subsequent works, we will attempt to replicate these findings in other PCa cohorts. 

## 5. Conclusions

In conclusion, our study has added validity to the notion that CNAs have a prognostic significance in PCa. We confirmed that gene level CNAs described in previous studies have prognostic values. We also identified new gene CNAs with potential prognostic values in PCa and described CNA marker panels that could impact risk stratification. Finally, this study defined gene CNA statuses that could predict disease progression in PCa.

## Figures and Tables

**Figure 1 genes-14-00956-f001:**
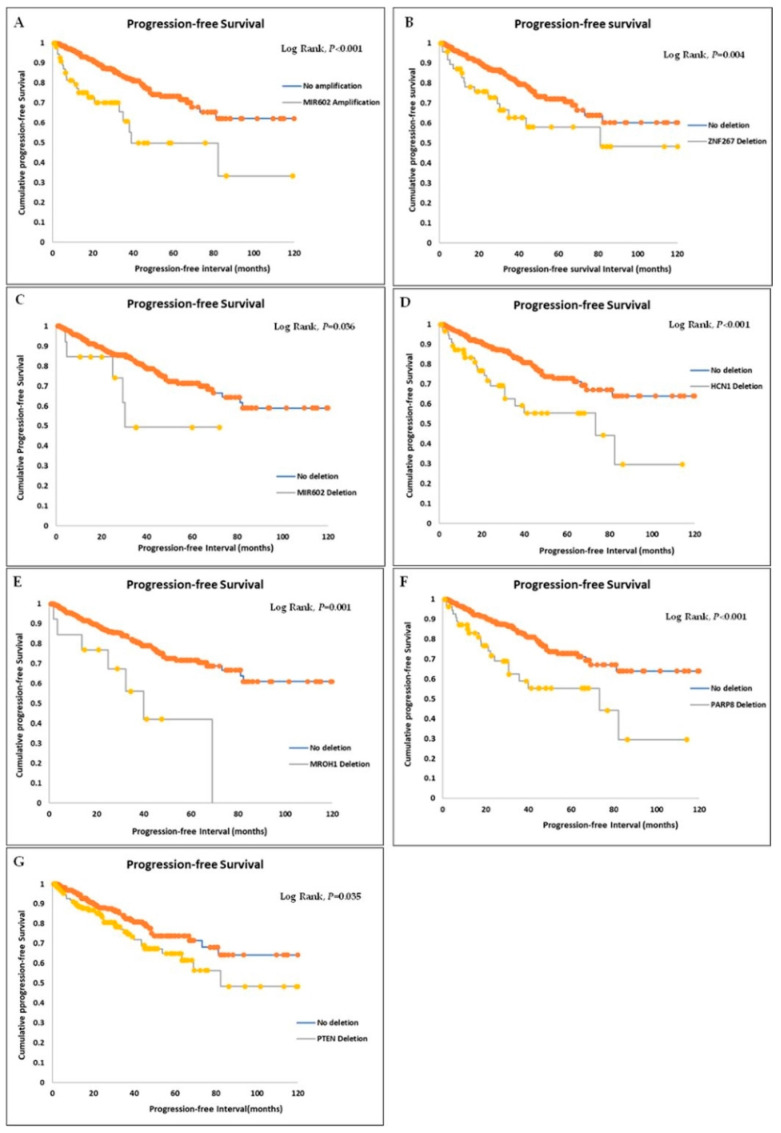
Association of CNAs of MIR602 (**A**,**C**), ZNF267 (**B**), HCN1 (**D**), MROH1 (**E**), PARP8 (**F**), and PTEN (**G**) with 120 month progression free survival.3.5. Predictive Panels of Genetic Markers Identified from Regression Analyses.

**Table 1 genes-14-00956-t001:** Clinical and molecular predictors of progression-free survival.

Clinical and Molecular Features	HR *	95.0% C.I. * HR	*p* Value
Disease stage	4.002	2.117–7.565	1.97 × 10^−5^
Prognostic group grade	0.609	0.401–0.926	2.05 × 10^−2^
*MIR602* deletion	2.795	1.044–7.484	4.09 × 10^−2^
*MIR602* amplification	1.917	1.089–3.376	2.42 × 10^−2^
*PARP8* deletion	1.724	1.024–2.904	4.05 × 10^−2^
*ZNF267* deletion	2.256	1.280–3.975	4.88 × 10^−2^
*MROH1* deletion	2.519	1.116–5.687	2.62 × 10^−2^
*HCN1* deletion	1.774	1.056–2.981	3.03 × 10^−2^

* HR = hazard ratio, C.I. = confidence interval.

**Table 2 genes-14-00956-t002:** Best Regression Model for PCa Stage Prediction.

Predictors	B	*p* Value	Exp(B)	95% C.I. Exp(B)
*SPOP* alteration	1.554	0.014	4.728	1.362–16.414
*SPP1* alteration	2.583	0.013	13.231	1.716–102.024
*CCND1* amplification	1.425	0.006	4.157	1.500–11.522
*CDKN1B* deletion	0.762	0.007	2.143	1.228–3.740
*NKX3.1* deletion	0.715	0.001	2.045	1.341–3.118
*PTEN* deletion	0.709	0.004	2.032	1.256–3.287
*PARP8* deletion	0.978	0.029	2.658	1.106–6.386
Constant	−0.624	<0.001	0.536	

## Data Availability

Publicly available datasets were analyzed in this study. The datasets were freely available and can be accessed at https://portal.gdc.cancer.gov/repository and https://www.cbioportal.org/study/summary?id=prad_tcga_pan_can_atlas_2018 as TCGA-PRAD and Prostate Adenocarcinoma (TCGA PanCancer Atlas), respectively.

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
