# Peer review of "Prognostic Values of Gene Copy Number Alterations in Prostate Cancer"

_genes, 2023, doi:10.3390/genes14050956_

Round 1
Reviewer 1 Report
The study from Alfahed el al. explored the TCGA PCa cohort and identified new gene copy number alterations potentially interesting for prognostic purposes. They support the notion that CNAs might have prognostic significance in PCa and they identified new gene CNAs with potential prognostic value. Their results, however, showed just a modest performance with respect to accuracy, sensitivity, specificity, and predictive for localized vs advanced disease. As a pilot study, is very interesting and could be a starting point for deeper studies. Perhaps, an initial patient stratification including, for instance, the histological subtype (AR deletions might account for more aggresive, castration resistant or neuroendocrine PC), or the developmemt of hormone-refractory disease, could be usefull. Considering Gleason score and T stage is not enough, since their real predictive value is limited.
Reviewer 2 Report
Dear authors,
I recommend that you conduct a final review of the document's formatting. Please include sections such as: "Data Availability Statement", "Institutional Review Board Statement", "Supplementary Materials", and so on. You can get details here: https://www.mdpi.com/journal/genes/instructions.
Below, I describe some recommendations that can be adopted to improve your manuscript.
MAJORS:
The abstract must be improved. You cited the acronymous PCa but did not explain the meaning (PCa: prostate cancer). Please include all definitions used for the first time in the abstract (as well as you did in the first line of the introduction section). Additionally, you could include one or two sentences contextualizing the problem at the beginning of the abstract.
Line 70 – The “Genomic Data Analyses” section does not contain the necessary steps to understand what the authors did.
Line 72 – What is a “Windows-based Ubuntu 20.04 environment”?
Line 105 – “Excel spreadsheet from Ubuntu 20.04”. Excel is not compatible with Linux.
Line 138 – “(see Supplementary Data - Genes from altered segments)”. Where can I find the supplementary data?
Additionally, why the authors did not include a data statement in the end of the manuscript? What about the other statements?
Line 231 – Figure 1 shows several typing errors. There are many hidden texts. The figure still has a formatting problem, such as missing axes or lack of description of legends and colours.
MINORS:
Line 43 – Standardize the way you enter citations. This line is different from the others.
MINORS (text):
Line 31 - “Currently, the risk prediction for individual cases is of high priority in PCa management.”
Change to:
“Currently, risk prediction for individual cases is highly prioritised in PCa management”.
Line 33 – “a substantial fraction of patients develop aggressive disease which requires radical therapy”
Change to: “substantial fraction of patients develops an aggressive disease requiring radical therapy”.
Line 43 – “Over the years many studies have”
Change to: “Over the years, many studies have”
Line 46 – “Many studies have shown that gene CNAs is more important than somatic mutations”
Change to: “Many studies have shown that gene CNAs are more important than somatic mutations”
Line 74 – “Genetic markers which have been shown from previous studies to have prognostic values were included in the study.”
Change to: “Genetic markers shown from previous studies to have prognostic values were included in the study.”
Line 152 – “significant association with disease”
Change to: “significant association the with disease”
(153 and 155) – “the disease”
Line 209 - “Each of the 51 genetic markers were input into the models as amplification, deletion, and alteration,”
Change to:
“Each of the 51 genetic markers was input into the models as amplification, deletion, and alteration,”
Line 261: “Williams et al in their”
Change to:
“Williams et al. in their”
Line 301: “In subsequent works we will attempt”
Change to:
“In subsequent works, we will attempt”
Round 2
Reviewer 2 Report
The authors affirm that they submitted the required data as supplemental material, but no document was submitted as supplemental material. Also, the "Data Availability Statement" section does not include the access ID of the data used in this study. This will turn hard the task of reproducing their experiments.
On line 142, they quote: "Clinical and molecular data were output in Excel Comma Separated Values file (.csv)". This is just a CSV (Comma Separated Values) file. It does not need to include that it was generated in Excel. Additionally, I think it is not relevant to cite Ubuntu in the following phrase.
Figure 1 must be on only one page.
Also, there is an apparent contradiction in the "Funding statement" section. On line 482, they say: "This publication received no any financial support." But on line 484: "This study is supported via funding from Prince Sattam bin Abdulaziz University project number (PSAU/2023/R/1444)". Please fix this.
